Plasma thrombin-activatable fibrinolysis inhibitor and the 1040C/T polymorphism are risk factors for diabetic kidney disease in Chinese patients with type 2 diabetes

Huang Qinghua 1 2
Feng Dujin 3
Pan Lianlian 4
Wang Huan 3
Wu Yan 5
Zhong Bin 6
Gong Jianguang 7
Lin Huijun 3 lab_lhj@126.com
Fei Xianming 3 feixianming@hmc.edu.cn
1 Key Laboratory of Endocrine Gland Diseases of Zhejiang Province , Hangzhou, Zhejiang , China
2 Geriatric Medicine Center, Department of Endocrinology, Zhejiang Provincial People’s Hospital (Affiliated People’s Hospital), Hangzhou Medical College , Hangzhou, Zhejiang , China
3 Laboratory Medicine Center, Department of Clinical Laboratory, Zhejiang Provincial People’s Hospital (Affiliated People’s Hospital), Hangzhou Medical College , Hangzhou, Zhejiang , China
4 Department of Laboratory Medicine, Sanmen People’s Hospital , Sanmen, Zhejiang , China
5 Department of Laboratory Medicine, Lin’an First People’s Hospital , Hangzhou, Zhejiang , China
6 Department of Laboratory Medicine, The Seventh Cixi Hospital of Ningbo , Cixi, Zhejiang , China
7 Laboratory of Kidney Disease, Zhejiang Provincial People’s Hospital (Affiliated People’s Hospital), Hangzhou Medical College , Hangzhou, Zhejiang , China
Singh Himanshu
Electronic publication date: 2023 Nov 14
Publication date: 2023
Volume: 11
Electronic Location ID: e16352
Received 2023 Jun 8; Accepted 2023 Oct 4
Copyright: © 2023 Huang et al.
Copyright year: 2023
Copyright holder: Huang et al.
License: This is an open access article distributed under the terms of the Creative Commons Attribution License, which permits unrestricted use, distribution, reproduction and adaptation in any medium and for any purpose provided that it is properly attributed. For attribution, the original author(s), title, publication source (PeerJ) and either DOI or URL of the article must be cited.
License URL: https://creativecommons.org/licenses/by/4.0/

Keywords: Diabetic microangiopathy, Diabetic kidney disease, Thrombin-activatable fibrinolysis inhibitor, 1040C/T polymorphism, Risk factor

Funding: Medicine and Health Science and Technology Project of Zhejiang Province, China 2021KY060, 2022KY523, 2023KY023 Natural Sciences Fundation of Zhejiang Province, China LY20H190003 Tradictional Chinese Medicine Science and Technology Project of Zhejiang Province, China 2023ZL263 This work was supported by the Medicine and Health Science and Technology Project of Zhejiang Province, China (Grant Number 2021KY060 to Xianming Fei, Grant Number 2022KY523 to Qinghua Huang, and Grant Number 2023KY023 to Jianguang Gong), the Natural Sciences Foundation of Zhejiang Province, China (Grant Number LY20H190003) to Huan Wang, and the Traditional Chinese Medicine Science and Technology Project of Zhejiang Province, China (Grant Number 2023ZL263) to Jianguang Gong. The funders had no role in study design, data collection and analysis, decision to publish, or preparation of the manuscript.

==============================
Background

Inflammatory and hemostatic disorders in diabetic microangiopathy (DMA) can be linked to thrombin-activatable fibrinolysis inhibitor (TAFI) and its own gene polymorphisms. Thus, the study aimed to investigate the associations of plasma TAFI and gene polymorphisms with DMA in Chinese patients with type 2 diabetes (T2D).

Methods

Plasma TAFI of 223 patients with T2D was measured, and the genotypes and alleles of the 1040C/T, 438G/A, and 505G/A polymorphisms of the TAFI gene were analyzed. A ROC curve was constructed to evaluate the identifying power of TAFI between patients with T2D and DMA, and logistic regression analysis was used to observe the correlation of plasma TAFI and gene polymorphisms with the risk for DMA.

Results

Plasma TAFI was higher in patients with DMA than in patients with only T2D (p < 0.05). TAFI exhibited the largest area under ROC in identifying diabetic kidney disease (DKD) from only T2D (0.763, 95% CI [0.674–0.853], p < 0.01), and adjusted multivariate analysis showed a high odds ratio (OR: 15.72, 95% CI [4.573–53.987], p < 0.001) for DKD. Higher frequencies of the CT genotype and T allele of the 1040C/T polymorphism were found in DKD compared with only T2D (respectively p < 0.05), and the CT genotype exhibited a high OR (1.623, 95% CI [1.173–2.710], p < 0.05) for DKD. DKD patients with the CT genotype had higher plasma TAFI levels, while T2D and DKD patients with CC/TT genotypes had lower plasma TAFI levels.

Conclusion

Plasma TAFI and the CT genotype and T allele of the 1040C/T polymorphism are independent risk factors for DKD in Chinese T2D patients.

Introduction

Type 2 diabetes (T2D) has become a public health problem worldwide, with a high prevalence in adults (463 million people, 9.3% of the population) in 2019 (Saputro et al., 2021). Diabetes increases the risk of mortality and morbidity and is associated with long-term complications (Singh et al., 2022). Microvascular complications have been identified as the leading diabetic complication, and can also increase the risk of all-cause and cardiovascular disease (CVD) mortality (Huang et al., 2023), which is a major cause of death in diabetic patients. Moreover, deaths from microvascular complications account for 20% to 50% of the total deaths in diabetic patients (Murillo & Fernandez, 2016). Therefore, early monitoring, effective evaluation, and timely diagnosis of the occurrence of diabetic microangiopathy (DMA) would be of importance to delay the progression of the disease and reduce the death risk of diabetic patients.

The major types of DMA include diabetic kidney disease (DKD), diabetic retinopathy (DR), and diabetic peripheral neuropathy (DPN), which can lead to kidney failure, blindness, and various peripheral sensory symptoms (Preguiça et al., 2020). In previous studies, a variety of biomarkers have exhibited high clinical values in risk prediction and prognostic assessment of different types of DMA to varying degrees (Cañadas-Garre et al., 2019; Pan et al., 2018; Fei et al., 2018; Yang et al., 2022; Soedarman et al., 2022; Weng et al., 2022; Chen et al., 2023). However, the direct relationship between these biomarkers and microvascular damage has not yet been completely clarified. Furthermore, these biomarkers can also be affected by many uncontrollable factors. Hence, it remains unclear whether they are involved in the process of DMA development as the regulatory factors. Moreover, these biomarkers are not also recognized as the risk factors of overall DMA. On the other hand, some studies of genetic polymorphisms revealed that paraoxonase-1 L55M and Q192R genetic polymorphisms play important roles in microangiopathy susceptibility (Wu et al., 2018), and superoxide dismutase-2 (SOD2) CC genotype gives antioxidant protection against DKD (Oikonomou et al., 2018), and other various genetic polymorphisms in DMA have also been reviewed (Lyssenko & Vaag, 2023), suggesting that some polymorphic genes have specific clinical value in DMA assessment. However, up until now, systematic evaluation of these biomarkers in distinct types of DMA has been insufficient. Moreover, it is difficult to early predict and screen out the occurrence of diabetic complications. Thus, exploration of specific biomarkers and the polymorphic genes for risk and progression assessment of the various types of DMA will be of great practical significance.

It is widely believed that the pathogenic mechanisms of DMA are related to inflammation, endothelial dysfunction, oxidative stress, and abnormal hemostasis (McKay et al., 2016). The occurrence and development of DMA are influenced by various objective factors such as duration of diabetes, glycemic control, blood pressure, and dyslipidemia, but the significant differences in clinical outcomes of different diabetic patients are more likely related to genetic factors (Kwak & Park, 2015). For example, the promoter polymorphism of CCR5 gene is related to the occurrence of DMA (Zhang et al., 2016). Many studies have shown that T2D patients with microvascular complications generally have chronic inflammation (Rübsam, Parikh & Fort, 2018; Hois et al., 2016; Daugherty et al., 2018), enhanced coagulation activity, and reduced anti-coagulating and fibrinolytic activity (Hori et al., 2002; Ebara et al., 2018; Yano et al., 2003; Wang et al., 2023), thus promoting the occurrence and progression of microvascular damage. Therefore, undergoing inflammation and hemostatic disorders likely play important roles in the regulatory mechanisms of DMA, implying that identification of relevant biomarkers that regulate the inflammatory response and coagulating conditions, as well as the analysis of genetic polymorphisms for the risk and progression of DMA will contribute to the early identification and the improvement of prognosis of the patients.

Thrombin-activatable fibrinolysis inhibitor (TAFI) is a single-chain glycoprotein mainly synthesized in the liver. It is a carboxypeptidase B precursor, and vascular endothelial cells can also synthesize and secrete TAFI (Plug & Meijers, 2016). TAFI can be activated by thrombin to become TAFIa with anti-fibrinolytic (Bajzar, 2000) and anti-inflammatory effects (Grosso et al., 2017). These activities suggest that TAFI can link the regulation of coagulation activation, fibrinolysis inhibition, and inflammatory regulation (Plug & Meijers, 2016). At present, the TAFI gene encoding and regulating TAFI expression exhibits different influences on blood TAFI levels and the activity (Wu & Xu, 2008). Three single-nucleotide polymorphisms (SNPs), 505G/A (rs3742264), 1040 C/T (rs1926447), and −438G/A (rs2146881), of the TAFI gene were found to have subtle associations with venous thrombosis risk in White individuals (Wang et al., 2016). However, there were few relevant studies of polymorphic genes in different types of DMA of any population, and whether plasma TAFI levels and SNPs of the gene are related to the risk of DMA in Chinese T2D patients is also unknown. Thus, we conducted this study to evaluate the associations of plasma TAFI levels and genetic polymorphisms of the TAFI gene with the risk of overall DMA and amongst the different types of DMA in Chinese T2D patients.

Materials and Methods

Study population

A total of 223 patients with T2D were enrolled after applying the inclusion and exclusion criteria of the study. The subjects were recruited from the Department of Endocrinology, Zhejiang Provincial People’s Hospital between January 2020 and December 2021, including 146 males and 77 females, aged 37–85 years. They included 53 patients with T2D only, 63 patients with DKD, 26 patients with DPN, and 81 patients with diabetic multi-microvascular complications (DMC). The diagnosis of T2D and DMA was made according to the criteria in Diabetes Guideline 2020 of the China Diabetes Association (2021). The inclusion criteria included: (1) presence of T2D; (2) presence of untreated DKD; (3) presence of untreated DPN; (4) presence of untreated DMC. The exclusion criteria were as follows: (1) presence of type 1 diabetes; (2) presence of thrombotic diseases; (3) presence of primary liver and kidney disorders; (4) a recent history of surgery; (5) presence of cardiovascular and cerebrovascular diseases; (6) diagnosis of cancer; (7) presence of acute infection. Written informed consent was obtained from all individual participants included in the study. This study was performed in line with the principles of the Declaration of Helsinki. Approval was granted by the ethical committee of Zhejiang Provincial People’s Hospital (Approval No. 2021KT014).

Sample collection and laboratory assays

Fasting venous blood of all subjects was collected in vaccutainer tubes with EDTA-K2 (2 mL blood), sodium citrate (2.7 mL blood), and anti-coagulant-free (5 mL blood) (BD Inc., Franklin Lakes, NJ, USA) before any treatment. The blood samples were centrifugated at 1,500 × g at room temperature for 5 and 10 min, respectively, then the sera and plasma were collected off the top fraction without disturbing the cell layer. After plasma was isolated, the leukocyte-enriched white cell pellet was aspirated and resuspended in 1 mL of elution buffer (Tsingke, Beijing, China) and stored at −80 °C, followed by total genomic DNA extraction. An automatic hematological analyzer (BC-7500; Mindray Inc., Shenzhen, China) was used to analyze the white blood cells (WBC) and the subtypes count, platelets count, and high sensitive C-reactive protein in blood from the EDTA-K2 tubes. Subsequently, the levels of serum biochemical markers and plasma coagulation parameters were measured by an AU5800 clinical chemistry analyzer (Beckman-coulter, Brea, CA, USA) and CN-9000 coagulation analyzer (Sysmex, Kobe, Japan), respectively, which included serum lipids, thyroid stimulating hormone (TSH), free triiodothyronine (FT3), and free thyroxine (FT4), plasma fibrinogen (FIB), d-dimer (D-D), and fibrin/fibrinogen degradation products (FDPs). After the measurements, the remaining plasma was stored at −80 °C for further analysis. At the same time, spot urine samples were also collected to measure the levels of urinary creatinine and albumin, which were analyzed by the AU5800 clinical chemistry analyzer (Beckman-Coulter, Brea, CA, USA). Finally, the urinary albumin to creatinine ratio was calculated.

ELISA detection

Frozen plasma was thawed in a 37 °C water bath. The levels of TAFI antigen, antithrombin (AT), and soluble thrombomodulin (sTM) were measured by an enzyme-linked immunosorbent assay (ELISA) (mlbio Co. Ltd., Shanghai, China), and a microplate reader (Bio-Rad, Hercules, CA, USA) was used to read the optical density (OD) values of the reaction wells at a wavelength of 450 nm to calculate the concentrations of all samples. Standard curves were constructed according to the concentrations and OD values of standard materials for TAFI, AT, and sTM. The concentrations of plasma TAFI, AT, and sTM in patient samples were obtained by reference to the standard curves by the software of the microplate reader.

DNA extraction and identification

After thawing the leukocytes-enriched fraction of blood at room temperature, the suspensions were mixed gently by inverting five times. A total of 100 μL of sample was pipetted into a deep-well plate (1.2 mL). Then a TrellefTM Animal Genomic DNA kit (Tsingke, Beijing, China) was used to extract the total genomic DNA of the leukocytes according to the manufacturer’s instructions. After sufficient elution, the DNA was separated by agarose gel electrophoresis (JY600C; Junyi, Beijing, China). Briefly, the DNA samples (2 μL sample + 6 μL bromophenol blue) were loaded into an agarose gel and separated by electrophoresis at a voltage of 300 V for 12 min to obtain an identification gel map, and to confirm whether the DNA extraction conditions were single or diffuse, and whether there were any specific bands. Subsequently, the DNA supernatant was transferred to a 96-well plate (0.2 mL/well) and stored at −20 °C for the following experiments.

Design of primers and probes

The target sequence of TAFI gene was obtained from the NCBI website. Then TaqMan probes were designed, and the same gene sequence and PCR primers were used for the detection of different genotypes of TAFI SNPs. The 5′ ends of the TaqMan probes were labeled with different fluorescent dyes, 6-carboxyfluorescein (FAM) and phosphoramidite (VIC), respectively. However, the minor groove binder (MGB) was attached to the 3′ end as a nonfluorescent quencher (NFQ). All oligonucleotides were synthesized by General Biological Systems Co., Ltd. (Hefei, Anhui, China). The sequences of specific primers and fluorescent probes for the TAFI gene are listed in Table 1.

Table 1 The primer and probe sequences.

Primers	Sequences	Modification	
5′-	3′-	
rs3742264-F	TGGCATGGATTCCACAGTCA			
rs3742264-R	CTTTTGGGGTTTCTTTGAGCAG			
rs3742264-T-FAM	CATTTTTGGCTGTTTGT	FAM	MGB	
rs3742264-C-VIC	CATTTTTGGCTGCTTG	VIC	MGB	
rs1926447-F	GCTTCCAGTCTCTAGTAGCCAGTGA			
rs1926447-R	AAATACTTACATAAGGTTTCTGAGCCAT			
rs1926447-C-FAM	TGCTATTGAGAAAAcTAG	FAM	MGB	
rs1926447-T-VIC	TGCTATTGAGAAAAtTAGT	VIC	MGB	
rs2146881-F	TTGGATTGGATTAGCCAGATTTG			
rs2146881-R	CGTGAAGTACTATAAATGTCTTCCCTT			
rs2146881-A-FAM	CTCACTTTTTACATCTTC	FAM	MGB	
rs2146881-G-VIC	CTCACTTTTTACGTCTTC	VIC	MGB	
Note:

MGB, minor groove binder; FAM, 6-carboxyfluorescein; VIC, phosphoramidite.

Analysis of the TAFI gene single-nucleotide polymorphisms

The extracted DNA products were used as a RT-PCR template and amplified with 2×T5 Fast qPCR Mix (TP001I-5, Probe). The components of the reaction mixture are as follows: 2×T5 Fast qPCR Mix (Probe) 10 μL, 10 μM forward primer 1 μL, 10 μM reverse primer 1 μL, template (cDNA) 1 μL, and ddH2O 7 μL. The DNA products were amplified according to the following program by a PCR thermocycler (A300; LongGene, Hangzhou, China): initial cycle at 95 °C for 1 min; 40 cycles at 95 °C for 15 s; 40 cycles at 90 °C for 30 s; fluorescence was detected at 60 °C.

Statistical analysis

Initially, the Kolmogorov–Smirnov test was used to analyze the distribution normality of the data, and they are presented as the mean ± standard deviation ( x¯±sd) and median when distributing normally and non-normally, respectively. The non-normal and normal distribution data of the patient’s characteristics were analyzed by Mann–Whitney U test and Student’s t-test, respectively. Categorical data (percentage) were analyzed by chi-square test. The odds ratios (OR) and 95% confidence intervals (95% CI) of all factors were calculated using univariate logistic regression analysis to find the risk factors, then an adjusted-multivatiate analysis was performed including the risk factors from univariate analysis to evaluate the OR values and 95% CI of TAFI for different DMA. The receiver operating characteristic (ROC) curves were established, and the area under the curve (AUC) was calculated to evaluate the identifying ability of the independent risk factors between T2D only and different DMA. The amount of same allele in each genotype of TAFI was counted, and the allele frequencies were estimated by gene-calculating in all alleles and departure from the Hardy–Weinberg equilibrium was tested using χ2 test with 1 degree of freedom. The statistical package SPSS 20.0 (SPSS; IBM Corp., Armonk, NY, USA) was used, and p-values of less than 0.05 (0.10 for regression analyses) were considered statistically significant.

Results

Basic characteristics and laboratory indicators of patients

The patients in the study were assigned to two groups including the overall DMA group (170 subjects) and the T2D only group (53 subjects). Of all the baseline characteristics, only age and duration of diabetes showed significant differences between the two groups (p < 0.05 and p < 0.01, respectively); however, the levels of serum TSH, plasma FIB, FDPs, AT, TAFI, and sTM, as well as the urinary ACR were significantly different between the two groups (p < 0.05 or p < 0.001, respectively). There were no statistical differences for other variables (all p > 0.05) (Table 2). Figure 1 shows plasma TAFI levels in the two groups, which were significantly higher in the overall DMA group compared to the T2D only group (p < 0.01), but there was no statistical differences in TAFI levels between the DKD, DPN, and DMC groups (all p > 0.05). Groups classified according to type of DMA still demonstrated higher levels of plasma TAFI than that of the patients having only T2D (data not presented, all p < 0.001).

Table 2 Comparisons of basic characteristics between the subjects in only T2D and overall DMA.

Variables	Group only T2D	Group overall DMA	Statistical value	p-value	
n	Values	n	Values	
Age (year)	53	53.17 ± 15.84	170	60.3 ± 13.3	3.233	0.001	
Sex (M, %)	53	35, 66.04	170	111, 65.68	0.010	0.921	
Smoking habit (n, %)	53	16, 30.19	170	54, 31.95	0.034	0.853	
Duration of diabetes (year)	53	5 (1–12)	170	10 (2–15)	3,661.000	0.039	
BMI (kg/m2)	53	25.89 ± 4.32	168	25.156 ± 3.90	1.067	0.287	
SBP (mmHg)	53	134.81 ± 15.80	170	138.896 ± 20.57	1.217	0.210	
DBP (mmHg)	53	80.48 ± 9.17	170	81.216 ± 11.92	0.473	0.637	
WBC (×109/L)	52	6.56 ± 1.61	169	6.88 ± 1.96	0.963	0.337	
LY (%)	52	30.05 ± 8.93	169	29.44 ± 9.14	4,103.500	0.471	
GR (%)	52	60.72 ± 8.54	169	60.87 ± 11.34	4,209.000	0.646	
PLT (×109/L)	52	220 (83–248)	169	207 (160–245)	4,018.500	0.352	
HS-CRP (mg/L)	49	1.40 (1.30–3.33)	165	1.65 (1.30–4.58)	3,529.000	0.164	
FIB (g/L)	53	2.55 (2.26–2.94)	170	2.84 (2.42–3.62)	3,261.500	0.002	
D-D (ng/mL)	53	240 (140–365)	170	300 (160–530)	3,791.000	0.082	
FDP (ng/mL)	53	1,185 (540–2,000)	170	2,000 (1,585–2,300)	2,765.000	0.000	
CHOL (mmol/L)	52	4.60 (3.62–5.51)	170	4.36 (3.57–5.19)	3,994.000	0.293	
LDL-C (mmol/L)	52	2.41 (1.92–3.29)	170	2.40 (1.90–3.06)	3,982.500	0.280	
TSH (U/L)	52	1.38 (0.90–2.07)	161	1.69 (1.08–2.57)	3,228.500	0.026	
FT3 (ng/L)	52	3.16 (2.96–3.39)	161	3.16 (2.83–3.53)	4,170.500	0.968	
FT4 (ng/dL)	52	9.56 (8.87–11.10)	161	9.41 (8.56–10.56)	3,830.500	0.358	
HbA1c (%)	50	9.50 (7.30–11.60)	164	10 (8.50–11.68)	3,705.000	0.303	
ACR (mg/g)	50	7.81 (1.13–8.00)	165	8.00 (2.07–19.23)	3,182.000	0.014	
AT (μg/mL)	53	6.78 ± 3.02	170	4.75 ± 2.22	5.288	0.000	
TAFI (μg/mL)	53	4.09 ± 0.92	170	5.18 ± 2.36	3.267	0.001	
sTM (μg/mL)	53	3.98 ± 2.25	170	5.23 ± 3.00	2.814	0.005	
Note:

Data were presented as mean and standard deviation, the median and range, and percent. T2D, type 2 diabetes; DKD, diabetice kidney disease; WBC, white blood cell; GR, granulocyte; HS-CRP, high sensitive C reactive protein; FIB, fibrinogen; D-D, D‐dimer; FDP, fibrin(ogen) degradation products; CHOL, cholesterol; LDL-C, low density lipoprotein-cholesterol; TSH, thyroid stimulating hormone; FT3, free triiodothyronine; FT4, free thyroxine; HbA1c, glycosylated hemoglobin; ACR, albumin-creatine ratio; AT, antithrombin; TAFI, thrombin-activatable fibrinolysis inhibitor; sTM, thrombomodulin. The statistical and p-values were analyzed by Student t-test, Mann-Whitney U test, and chi-square test, respectively.

Figure 1 Plasma TAFI levels in patients with only T2D and different types of DMA.

DMA, diabetic microangiopathy; DKD, diabetic kidney disease; DPN, diabetic peripheral neuropathy; DMC, diabetic multi-microvascular disease; T2D, type 2 diabetes; TAFI, thrombin-activated fibrinolysis inhibitor. (A) Comparison between pure T2D and overall DMA analyzed by student-t test; (B) Comparisons among DKD, DPN and DMC analyzed by one-way analysis of variance.

Univariate regression and ROC curve analysis

Inflammatory indicators (WBC and granulocyte counts) and the major variables with statistical differences (age, duration of diabetes, FIB, FDP, AT, sTM, and TAFI) between the overall DMA group and T2D only group were included to perform a univariate regression analysis for the risk of overall DMA, DKD, DPN, and DMC, respectively. We found that the variables all showed significant differences for risk assessment of the different types of DMA, but only plasma TAFI and AT were the common risk factors for overall DMA and the different types of DMA (p < 0.05 and p < 0.01, respectively) (Table 3). Therefore, we further examined the association of plasma TAFI between patients with different types of DMA and those with only T2D. ROC curve analysis showed that plasma TAFI had the largest AUC (0.763, 95% CI [0.674–0.853]) in identifying DKD patients from T2D only patients with a sensitivity of 0.571 and specificity of 0.924 at an optimal cutoff value of 4.99 μg/mL. The second was for DPN (0.716, 95% CI [0.594–0.837]), and the AUC for DMC was the smallest (Fig. 2, Table 4).

Table 3 Univariate logistic regression analyse of risk factors for different types of DMA.

Variables	FOR overall DMA	For DKD	For DPN	For DMC	p-value	
OR (95% CI)	p-value	OR (95% CI)	p-value	OR (95% CI)	p-value	OR (95% CI)	
Age	1.037 [1.014–1.061]	0.002	1.035 [1.008–1.062]	0.009	1.021 [0.988–1.054]	0.217	1.038 [1.012–1.065]	0.004	
Duration of diabetes	1.041 [1.012–1.086]	0.012	1.026 [0.979–1.075]	0.279	0.996 [0.931–1.065]	0.897	1.038 [1.014–1.109]	0.01	
WBC	1.089 [0.916–1.294]	0.335	1.315 [1.49–1.649]	0.018	1.081 [0.825–1.415]	0.572	0.940 [0.759–1.163]	0.568	
GR	1.007 [0.96–1.029]	0.678	1.036 [0.992–1.081]	0.106	0.961 [0.913–1.010]	0.117	1.005 [0.969–1.043]	0.769	
FIB	1.850 [1.198–2.856]	0.006	2.689 [1.449–4.990]	0.002	1.520 [0.909–2.541]	0.111	1.592 [1.015–2.497]	0.043	
FDP	1.000 [1.000–1.000]	0.117	1.000 [1.000–1.000]	0.067	1.000 [1.000–1.000]	0.741	1.000 [1.000–1.000]	0.317	
AT	0.739 [0.642–0.851]	0.000	0.727 [0.628–0.870]	0.000	0.646 [0.460–0.909]	0.012	0.640 [0.513–0.799]	0.000	
TAFI	1.522 [1.177–1.970]	0.001	2.073 [1.423–3.019]	0.000	1.787 [1.100–2.903]	0.019	1.340 [1.044–1.718]	0.021	
sTM	1.296 [1.014–1.061]	0.002	1.131 [0.955–1.3380]	0.153	2.277 [1.562–3.319]	0.000	1.232 [0.994–1.527]	0.056	
Note:

DMA, diabetic microangiopathy; DKD, diabetic kidney disease; DPN, diabetic peripheral neuropathy; DMC, diabetic multi-microvascular disease; WBC, white blood cell; GR, granulocyte; FDP, fibrin(ogen) degaration products; AT, antithrombin; TAFI, thrombin-activatable fibrinolysis inhibitor; sTM, thrombomodulin; OR, odds ratio; CI, confidence interval. P-value, analyzed by a univariate logistic regression analyse including pure type 2 diabetes and different DMA.

Figure 2 ROC curves of plasma TAFI levels in discriminating between only T2D and different DMA.

ROC, receiver operating characteristic; DMA, diabetic microangiopathy; DKD, diabetic kidney disease; DPN, diabetic peripheral neuropathy; DMC, diabetic multi-microvascular disease.

Table 4 Identifying ability of plasma TAFI for different types of DMA from only T2D by ROC curve analysis.

Type of DMA	AUC	Cutoff-value (μg/mL)	Sensitivity	Specificity	NPV	PPV	Youden index	Coincidence rate (%)	
Overall DMA	0.663 (0.590–0.735)	4.99	0.459	0.924	0.37	0.86	0.38	69	
DKD	0.763 (0.674–0.853)	4.99	0.571	0.924	0.32	0.88	0.50	75	
DPN	0.716 (0.594–0.837)	4.48	0.577	0.773	0.35	0.72	0.35	68	
DMC	0.568 (0.471–0.664)	5.01	0.395	0.925	0.39	0.84	0.32	66	
Note:

DMA, diabetic microangiopathy; T2D, type 2 diabetes; DKD, diabetic kidney disease; DPN, diabetic peripheral neuropathy; DMC, diabetic multi-microvascular disease; TAFI, thrombin-activatable fibrinolysis inhibitor; AUC, area under curve; ROC, receiver operating characteristic; NPV, negative predicting value; PPV, positive predicting value.

Adjusted-multivariate regression analysis

Plasma TAFI was first treated as a continuous variable, then it was further treated as a categorical variable expressed as “increased” or “decreased” by using the optimal cutoff values from ROC curve analyse. When including other risk factors (confounders) from univariate regression analyse, the adjusted-multivariate regression analysis indicated that plasma TAFI was an independent risk factor (with high ORs) not only for overall DMA, but for DKD and DMC as well (p < 0.05 or <0.001 respectively). However, plasma TAFI level did not exhibit the statistically significant ORs for DPN (both p > 0.05) (Table 5). Furthermore, when all DMA patients were divided into four groups (Group Q1: <3.62 μg/mL; Group Q2: 3.62–4.44 μg/mL; Group Q3: 4.44–5.49 μg/mL; Group Q4: >5.49 μg/mL) based on the quartiles of plasma TAFI levels, the incidence of overall DMA was 59.6%, 68.5%, 82.1%, and 94.6% in the 1st, 2nd, 3rd, and 4th quartile (Q1, Q2, Q3 and Q4), respectively (p-trend < 0.01) (Fig. 3).

Table 5 Adjusted-multivariate regression analyse of plasma TAFI levels for risk of different types of DMA.

Type of DMA	TAFI as continuous variable	TAFI as categorical variable	
OR	95% CI	p-value	OR	95% CI	p-value	
DMA	2.120	[1.461–3.076]	0.000	6.542	[2.445–17.522]	0.000	
DKD	2.910	[1.719–4.926]	0.000	15.712	[4.573–53.987]	0.000	
DPN	1.815	[0.752–4.380]	0.185	2.410	[0.532–10.908]	0.254	
DMC	3.681	[2.018–6.714]	0.000	3.663	[1.325–10.125]	0.012	
Note:

Confounders for regression analyse were as follows: DMA (age, duration of diabetes, Fbg, AT-III, and sTM); DKD (age, duration of diabetes, WBC, Fbg, FDP, and AT-III); DPN (Age, duration of diabetes, AT-III, and sTM); DMC (age, duration of diabetes, Fbg, AT-III, and sTM). Categorical variables were classified based on the optimal cut-off values of TAFI in each group. DMA, diabetic microvascular complication; DKD, diabetic kedney disease; DPN, diabetic peripheral neuropathy; DMC, diabetic multi-complications; OR, odds ratio; CI, confidence interval ; TAFI, thrombin-activatable fibrinolytic inhibitor. P-value was analyzed by logistic regression analysis by adjusted the potential confounders.

Figure 3 Incidence of overall DMA in different quartile of plasma TAFI levels.

Group Q1: <3.62 μg/mL, Group Q2: 3.62–4.44 μg/mL, Group Q3: 4.44–5.49 μg/mL, group Q4: ≥5.49 μg/mL. DMA, diabetic microangiopathy; TAFI, thrombin-activated fibrinolysis inhibitor. trend-p < 0.01 analyzed by trend Chi-square test; *p < 0.05, and **p < 0.01 compared with group Q2 analyzed by chi-square test, respectively.

Analysis of the SNPs and genotype of the TAFI gene

The theoretical allele frequencies of the genetic polymorphisms in this study can be calculated by using the investigated frequencies. Then Chi-square test was performed, and no deviation from the Hardy–Weinberg equilibrium was observed (p > 0.05, data not shown). Table 6 presents the results of the association analysis between the SNPs of the TAFI gene and the risk for different types of DMA. In the results, the CT genotype of the 1040C/T polymorphism was found to be a risk factor, and the T allele showed a higher frequency in patients with DKD than those with only T2D (41.3% vs. 13.3% and 20.6% vs. 10.0%, respectively, both p < 0.05). Moreover, DKD patients exhibited a lower percentage of the GA genotype and lower frequency of the A allele of the 505G/A polymorphism, than that of patients with T2D only (both p < 0.05). Further regression analysis revealed that the CT genotype of 1040C/T polymorphism was the only independent risk gene of DKD (OR: 1.623, 95% CI [1.173–2.710], p < 0.05). Other TAFI gene polymorphisms examined in this study did not indicate a close association with any other type of DMA (p > 0.05).

Table 6 Distribution of the genotypes and alleles of TAFI gene polymorphisms in patients with only T2D and different types of DMA.

SNP	Genotype/allele	n	Pure T2D (n,%)	DMA (n, %)	OR (95% CI)	p-value	χ2	p-value	DKD (n, %)	OR (95% CI)	p-value	χ2	p-value	
(A)	
rs1926447
1040C/T	TT	5	1 (3.33)	4 (0.87)					2 (3.17)					
CT	44	4 (13.3)	40 (34.8)	1.353 [0.831–2.202]	0.244	5.285	0.071	26 (41.3)	1.623 [1.173–2.710]	0.044	7.349	0.025	
CC	96	25 (83.3)	71 (61.7)					35 (61.9)					
T	54	6 (10.0)	48 (20.9)			3.710	0.054	30 (20.6)			4.966	0.026	
C	236	54 (90.0)	182 (79.1)					96 (79.4)					
rs2146881
438G/A	AA	4	0 (0.00)	4 (3.5)					2 (3.17)					
GA	35	7 (23.3)	28 (24.3)	1.387 [0.592–3.248]	0.451	1.119	0.571	18 (28.6)	1.621 [0.641–4.097]	0.307	0.509	0.775	
GG	106	23 (76.7)	83 (72.2)					43 (68.3)					
A	43	7 (11.7)	36 (15.7)			0.370	0.543	22 (17.5)			0.462	0.497	
G	247	53 (88.3)	194 (84.3)					104 (82.5)					
rs3742264
505G/A	GG	87	13 (53.3)	74 (64.3)					44 (69.8)					
GA	48	15 (50.0)	33 (28.7)	1.314 [0.650–2.660]	0.447	5.016	0.081	16 (25.4)	1.307 [0.821–2.080]	0.260	6.053	0.048	
AA	10	2 (6.67)	8 (6.95)					4 (6.3)					
G	222	41 (68.3)	181 (78.7)			2.271	0.132	104 (82.5)			4.774	0.029	
A	68	19 (31.7)	52 (21.3)					22 (17.5)					
(B)	
rs1926447
1040C/T	TT		1 (3.33)	1 (4.55)					1 (3.33)					
CT		4 (13.3)	7 (31.8)	1.390 [0.736–2.623]	0.310	2.755	0.252	7 (23.3)	1.103 [0.597–2.038]	0.755	1.010	0.604	
CC		2 5 (83.3)	14 (63.6)					22 (73.4)					
T		6 (10.0)	9 (20.5)			2.248	0.134	9 (15.0)			0.809	0.368	
C		54 (90.0)	35 (79.5)					49 (85.0)					
rs2146881
438G/A	AA		0 (0.00)											
GA		7 (23.3)	1 (4.55)	2.013 [0.659–6.153]	0.220	2.006	0.367	1 (3.33)	0.705 [0.218–2.279]	0.560	2.784	0.249	
GG		23 (76.7)	7 (31.8)					3 (10.0)					
A		7 (11.7)	14 (63.6)			1.506	0.220	26 (86.7)			0.370	0.543	
G		53 (88.3)	9 (20.5)					5 (8.33)					
rs3742264
505G/A	GG		13 (53.3)	35 (79.5)					55 (91.7)					
GA		15 (50.0)		1.804 [0.680–4.781]	0.236	4.007	0.135		0.912 [0.392–2.119]	0.560	0.286	0.867	
AA		2 (6.67)	15 (68.2)					15 (50.0)					
G		41 (68.3)	5 (22.7)			1.622	0.203	13 (43.3)			0.159	0.690	
A		19 (31.7)	2 (9.09)					2 (6.67)					
Note:

DMA, diabetic microangiopathy; T2D, type 2 diabetes; DKD, diabetic kidney disease; DPN, diabetic peripheral neuropathy; DMC, diabetic multi-microvascular disease; TAFI, thrombin-activatable fbrinolysis inhibitor; SNPs, single nucleotide polymorphisms; OR, odds ratio; CI, conûdence interval. OR and relative p-value: analyzed by risk regression analysis; χ2 and relative p-value: analyzed by Chi-square test among three genotypes and between different alleles of SNP.

Effect of TAFI gene haplotypes on TAFI level

According to the genotypes and alleles of the 1040C/T SNPs, plasma TAFI levels were compared among DKD patients with CC/TT and CT genotypes, and among T2D patients with the CC/TT genotype. DKD patients with CC/TT genotype showed a lower level of plasma TAFI than that with a CT genotype (4.74 (4.06–5.82) μg/mL vs. 5.65 (4.71–6.74) μg/mL; p < 0.05), but they had a higher plasma TAFI level than that of the T2D only subjects with a CC/TT genotype (4.74 (4.06–5.82) μg/mL vs. 3.86 (3.34–4.29) μg/mL; p < 0.05). Because of the small number of T2D subjects with the CT genotype (n = 4), data of the serum TAFI levels were not presented. Detailed information is shown in Fig. 4.

Figure 4 Plasma TAFI levels of only T2D and DKD with different genotypes of 1040C/T SNP.

T2D, type 2 diabetes; DKD, diabetice kidney disease; TAFI, thrombin-activatable fibrinolysis inhibitor. T2D-CC/TT, type 2 diabetes with CC or TT genotype; DKD-CC/TT, diabetic kidney disease with CC or TT genotype; DKD-CT, diabetic kidney disease with CT genotype; p-value was analyzed by Mann-Whitney U test.

Discussion

TAFI possesses powerful anti-inflammatory and anti-fibrinolytic activities, and thrombomodulin can enhance the power of TAFI activation (Yüzbaşıoğlu et al., 2019). Therefore, TAFI is likely to be a key regulator of the processes of inflammation, coagulation, and fibrinolysis. Previous studies have shown that the major pathophysiological mechanisms of DMA are local chronic inflammatory response and the subsequent abnormalities in anti-coagulation and anti-fibrinolytic pathways caused by microvascular endothelial cell damage and release of coagulation and fibrinolytic factors (Madonna et al., 2017; Camera, Hopps & Caimi, 2007; Gastaldi et al., 2021). Thus, TAFI may be a potential biomarker for DMA. In the present study, we found some obvious differences in blood biomarkers related to disorders of blood vessels, anti-coagulation, and fibrinolysis in DMA patients compared to patients with T2D only. These included elevated plasma TAFI, sTM, D-D, and FDP concentrations, and low AT activity, which may be associated with metabolic disorders in patients with DMA (Zhang et al., 2014). Oveall, DMA patients showed higher urinary ACR level, which is because that DMA patients also contained some DKD ones with undergoing albuminuria. Looking more deeply into the results, we found that plasma TAFI showed a high OR for overall DMA, and it also had a high OR for DKD, DPN, and DMC, suggesting that plasma TAFI is a risk factor for overall DMA, and that the increment of plasma TAFI level might be consistent with the increased risk for various types of DMA. Our study revealed probable blood vessel injury, enhanced anti-coagulation, and decreased fibrinolytic activity in DMA patients with increased plasma TAFI, which implies that plasma TAFI can be used as a potential biomarker of DMA risk.

In T2D progression, some biomarkers that directly reflect the disease state, may also be good predictors of DMA (Ziegler et al., 2022; Xie, Lin & Wang, 2019; Wu, Wu & Zhong, 2018). The previous study have shown that high plasma TAFI level is associated with an increased risk for T2D, and it may thus serve as a potential marker for the diagnosis of T2D (Zheng et al., 2015). Therefore, we speculated that plasma TAFI may be a good biomarker for the identification of overall DMA as well. In this study, TAFI did not exhibit a high identifying ability between overall DMA and T2D, but we found that TAFI was more likely to predict DKD. Moreover, we obtained the optimal cut-off points of plasma TAFI for different types of DMA from ROC curves, and when the cut-off point was set at 4.99 μg/mL, a lower sensitivity (0.57) and high specificity (0.92) in discriminating between DKD and T2D was observed, and TAFI exhibited the highest overall coincidence rate (0.75) in identifying DKD from other types of DMA. Therefore, the results suggest that plasma TAFI is a better predictor for DKD than any other DMA in our study, and it can be used as a specific biomarker for identifying DKD in T2D patients. When TAFI level was treated as a categorical variable according to the optimal cut-off value, our findings showed high ORs (17.362, 10.329 and 13.577) for overall DMA, DKD, and DMC, respectively. Consistent with these results, we also observed an increased incidence of overall DMA with the increment of plasma TAFI concentrations, and we speculated that the extreme differences of plasma TAFI levels between these subjects with overall DMA and T2D only might be a major contributor to development of DMA. In recent years, few studies have evaluated the significance of plasma TAFI in diabetes and DMA because it is not commonly used to predict and assess risk in diabetic subjects. The study concluded that TAFI may be participating in the mechanism of hypofibrinolysis and the occurrence of microvascular complications in diabetes (Sherif et al., 2014). Similarly, the present study suggests that the plasma TAFI is an independent risk factor for DKD and DMC.

There is a necessity to investigate the mechanisms that result in increased plasma TAFI and how they may be associated with DMA and its progression. It is known that TAFI is mainly expressed in the liver and released into the blood, but the transcription of TAFI has also been observed in fatty tissue of T2D patients, vascular endothelial cells, human peripheral blood mononuclear cells, and megakaryocytes and platelets (Plug & Meijers, 2016; Yano et al., 2003). Diabetes may result in an increase in pro-coagulation mechanisms and a decrease in anti-coagulation and fibrinolysis, which may lead to an elevated risk of thrombosis (Westein, Hoefer & Calkin, 2017; Liu et al., 2023). Therefore, high blood TAFI levels are likely accompanied by the development of diabetes, which is consistent with the low levels of fibrinolysis observed in diabetes. It has also been recognized that diabetes and its complications are associated with a series of changes in endothelial dysfunction caused by several factors including an excess of plasma FFAs, alterations in glucose metabolism, impaired insulin signaling, chronic inflammation, and oxidative stress (Sobczak & Stewart, 2019). These factors may cause blood to be in a hypercoagulable state, and promote the release of TAFI from the liver, platelets, vascular endothelial cells, fat cells, and inflammatory cells, which may be the main cause of increased TAFI and its close association with different types of DMA in T2D patients.

Although previous studies seem to indicate that TAFI gene polymorphisms are highly associated with circulating TAFI levels, and can also influence the TAFI level variability (Frère et al., 2005), some studies still indicate that less than 25% of the variation in plasma TAFI levels is due to TAFI gene polymorphisms in the non-encoding region that may modulate gene expression or affect mRNA stability (Sillen & Declerck, 2021). The association of TAFI gene polymorphisms with plasma TAFI level and DMA risk remain unclear in Chinese T2D patients, and it needs to be investigated. In a recent study, three important SNPs (505A/G, 1040C/T, and −438A/G) were investigated in association with venous thrombosis risk (Xu et al., 2012; Wang et al., 2016; Arauz et al., 2018), and TAFI polymorphisms (505G/A and 1040C/T) and circulating TAFI levels were analyzed in Chinese patients with T2D and macrovascular diseases (Zheng et al., 2015). However, the associations of the three SNPs with different types of DMA are still not clear. Therefore, in this study, we analyzed the genotypes and relative alleles of the three SNPs in overall DMA, DKD, and DMC. In the results, no SNP was significantly correlated with risk of overall DMA or DPN and DMC, but the 1040C/T (rs1926447) polymorphism showed a close association with the incidence of DKD in T2D patients. Moreover, in subgroup analyses, our study exhibited that the T allele was of a high frequency in DKD patients, which indicated that the T allele and the CT genotype of the rs1926447 polymorphism may increase the risk of DKD, and preventative interventions should be adopted for this population. Xu et al. (2012) found a lower frequency of T allele in diabetic nephropathy, which was not consistent with our results. One of the main causes for this inconsistency may be the difference of study populations. Investigating further, we did not find that any genotype or allele of the 505A/G and −438A/G polymorphisms were risk predictors of DKD, and we speculate that they may not be associated with DKD risk. In patients with T2D only and those with DKD, we compared the levels of plasma TAFI in different genotypes of the 1040C/T polymorphism. Higher level of plasma TAFI was observed in DKD patients with a CT genotype compared to that with CC/TT genotypes, but T2D only subjects with CC/TT genotypes exhibited a lower level of plasma TAFI compared to the DKD patients. These findings indicate that the CT genotype of the 1040C/T polymorphism is a risk factor for DKD, suggesting it may regulate the high expression of plasma TAFI to increase the risk of DKD. Our study implies that early intervention for T2D patients with the CT genotype of the 1040C/T polymorphism will be of importance for the prevention of DKD and prognostic improvement of T2D patients.

Within our knowledge, there are two limitations in this study. First, DKD was defined by clinical diagnosis based on declined renal function or proteinuria, and some patients with early stage DKD that would require diagnosis by histology and so were otherwise missed, were likely included in the T2D only group. Therefore, the identifying ability of plasma TAFI for DKD may have been reduced, leading to some uncertainty in the results. Second, as most patients with DR are outpatients, it was difficult to enroll enough DR patients in this study, so the overall DMA subjects in the present study did not include these patients. Therefore, we do not know whether the three SNPs can influence the plasma TAFI levels of DR patients and whether they are associated with DR risk. Despite the limitations, our study also revealed that increased plasma TAFI level and the T allele and CT genotype of the 1040C/T polymorphism were strongly correlated with DKD.

Conclusions

This study revealed that increased level of plasma TAFI and the CT genotype and T allele of the 1040C/T polymorphismare are risk factors for Chinese DKD patients, making the TAFI gene a possible risk gene for DKD. However, further multi-center studies with more diverse T2D patients are needed to draw more definitive conclusions.

Supplemental Information

Supplemental Information 1 Raw data.

Click here for additional data file.

Additional Information and Declarations

Competing Interests

Author Contributions

Human Ethics

DNA Deposition

Data Availability

The authors declare that they have no competing interests.

Qinghua Huang conceived and designed the experiments, performed the experiments, analyzed the data, prepared figures and/or tables, authored or reviewed drafts of the article, and approved the final draft.

Dujin Feng performed the experiments, analyzed the data, prepared figures and/or tables, authored or reviewed drafts of the article, and approved the final draft.

Lianlian Pan performed the experiments, analyzed the data, authored or reviewed drafts of the article, contributed data and materials management, and approved the final draft.

Huan Wang analyzed the data, prepared figures and/or tables, and approved the final draft.

Yan Wu performed the experiments, analyzed the data, authored or reviewed drafts of the article, and approved the final draft.

Bin Zhong performed the experiments, authored or reviewed drafts of the article, and approved the final draft.

Jianguang Gong performed the experiments, authored or reviewed drafts of the article, contributed reagents, materials, and approved the final draft.

Huijun Lin conceived and designed the experiments, analyzed the data, authored or reviewed drafts of the article, and approved the final draft.

Xianming Fei conceived and designed the experiments, authored or reviewed drafts of the article, and approved the final draft.

The following information was supplied relating to ethical approvals (i.e., approving body and any reference numbers):

The ethical committee of Zhejiang Provincial People’s Hospital.

The following information was supplied regarding the deposition of DNA sequences:

The sequences of specific primers and fluorescent probes for the TAFI gene are available in Table 1.

The following information was supplied regarding data availability:

The raw data are available in the Supplemental File.

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
