# Peer review of "Plasma thrombin-activatable fibrinolysis inhibitor and the 1040C/T polymorphism are risk factors for diabetic kidney disease in Chinese patients with type 2 diabetes"

_PeerJ, doi:10.7717/peerj.16352_

## Round 0.1 · original submission · Minor Revisions

Both reviewers seem to acknowledge the well-structured and executed nature of the manuscript. However, they have highlighted the need for improvements in the language quality and accuracy of the content. Additionally, there is a unanimous call for updating the references to encompass the year 2023. Specific points of concern are the missing information about OD reading, clarification of technical terms, elaboration on certain aspects of Table 2, and the need for more coherent conclusions across the manuscript.

Addressing these suggestions should enhance the overall quality and comprehensibility of the manuscript, leading to a more robust and impactful presentation of the research findings.

**Language Note:** The review process has identified that the English language must be improved. PeerJ can provide language editing services - please contact us at copyediting@peerj.com for pricing (be sure to provide your manuscript number and title). Alternatively, you should make your own arrangements to improve the language quality and provide details in your response letter. – PeerJ Staff

Reviewer 1 ·

Basic reporting

1. Manuscipt is well designed, executed and easy to understand.

2. The English language could be further improved. Typing error should be avoided. For example, in Sample Collection line "then the sera and plasma were collected off the top fraction without disturbing the cell layer." is repeated twice.

3. The article structure is ok. Raw data shared.

4. July 2023 is almost ending, but no references of 2023 were added. Literature should be updated.

Experimental design

well designed and executed.
Appropriate statistical methods were used.

Validity of the findings

results are described well.
ROC curve graphs should be provided.
The conclusion of abstract and main text should be matching.

·

Basic reporting

The article is well written. The language of the manuscript needs to improve a bit and also can be updated as no reference of 2023 is mentioned.

Experimental design

The method part is well-designed. However, I have a few comments
1. In line 167, at which OD reading was taken is missing.
2. Explain regression and allele frequency analysis.

Validity of the findings

1. In table 2 ACR also significantly differs so also explain about it
2. In table 2 abbreviation of FIB is missing
3. In line 246-249, improve sentence language.
4. In 244, did not mention about DPN
5. In line 250, Furthermore………… add about four groups
6. In line 251-252, not clear about quartiles (Q1……Q4). Explain a bit more about it and add figure related to it.
7. In line 255, add a reference.

---

## Round 0.2 · accepted · Accept

Based on the feedback from both reviewers, it appears that the revised manuscript has addressed the concerns raised in the initial review. Reviewer 1 has provided positive feedback on various aspects of the manuscript, including the incorporation of changes, the experimental design, and the interpretation of results. Reviewer 2, while not offering specific comments, does not raise any additional concerns.

Regarding the suggestions from Reviewer 2 about correcting the references in lines 104 and 302, these are minor issues that can be easily rectified.

Therefore, I recommend accepting the manuscript for publication with minor revisions. The authors should make the suggested corrections to the references, as pointed out by Reviewer 2, before finalizing the manuscript for publication.

Reviewer 1 ·

Basic reporting

The authors have incorporated all changes in the revised manuscript. The manuscript encompasses all parameters for a well written manuscript.

Experimental design

Hypothesis was genuine and well worked.
the methodology was well designed and executed.

Validity of the findings

results are well written and interpreted.
the ROC curves have been added.
conclusion is revised.

·

Basic reporting

no comment

Experimental design

no comment

Validity of the findings

no comment

Additional comments

I have few suggestions
In line 104, it should be Hois R, et al., (in et al. t is missing)
In line 302, Gastaldi G et al., 2022 (year is different than Reference 35). need to check it.